# The *Han:SPRD* Rat: A Preclinical Model of Polycystic Kidney Disease

**DOI:** 10.3390/biomedicines12020362

**Published:** 2024-02-03

**Authors:** Ioannis Kofotolios, Michael J. Bonios, Markos Adamopoulos, Iordanis Mourouzis, Gerasimos Filippatos, John N. Boletis, Smaragdi Marinaki, Manolis Mavroidis

**Affiliations:** 1Clinic of Nephrology and Renal Tranplantation, Laiko Hospital, Medical School, National and Kapodistrian University of Athens, 11527 Athens, Greece; 2Center of Basic Research, Biomedical Research Foundation, Academy of Athens, 11527 Athens, Greeceemavroeid@bioacademy.gr (M.M.); 3Heart Failure and Transplant Unit, Onassis Cardiac Surgery Center, 17674 Athens, Greece; bo_mic@yahoo.com; 4Department of Pharmacology, National and Kapodistrian University of Athens, 11527 Athens, Greece; imour@med.uoa.gr; 5Department of Cardiology, Attikon University Hospital, Medical School, National and Kapodistrian University of Athens, 12462 Athens, Greece

**Keywords:** polycystic kidney disease, ciliopathies, nephronopthisis, *Han:SPRD* rat, *PKD/Mhm* rat, *ADPKD*, anks6, PKDR1

## Abstract

Autosomal Dominant Polycystic Kidney Disease (ADPKD) stands as the most prevalent hereditary renal disorder in humans, ultimately culminating in end-stage kidney disease. Animal models carrying mutations associated with polycystic kidney disease have played an important role in the advancement of ADPKD research. The *Han:SPRD* rat model, carrying an R823W mutation in the *Anks6* gene, is characterized by cyst formation and kidney enlargement. The mutated protein, named Samcystin, is localized in cilia of tubular epithelial cells and seems to be involved in cystogenesis. The homozygous *Anks6* mutation leads to end-stage renal disease and death, making it a critical factor in kidney development and function. This review explores the utility of the *Han:SPRD* rat model, highlighting its phenotypic similarity to human ADPKD. Specifically, we discuss its role in preclinical trials and its importance for investigating the pathogenesis of the disease and developing new therapeutic approaches.

## 1. Introduction

Cystic kidney diseases are multisystemic disorders that manifest in both children and adults and arise from both genetic and non-genetic causes, which often lead to end-stage renal disease (ESRD) [1,2]. Genetic disorders include autosomal dominant and recessive polycystic kidney disease (ADPKD, ARPKD), nephronopthisis, von Hippel–Lindau, and tuberous sclerosis [3]. These inherited cystic kidney diseases are caused by disorders of the cilia, often referred to as ciliopathies. 

ADPKD is a highly prevalent heterogeneous disease, affecting 1 in 500 to 1000 people worldwide [4,5]. Approximately 10% of all patients receiving renal replacement therapy in Europe suffer from ADPKD [6,7]. Mutations in the *PKD1* (MIM# 601313) and *PKD2* (MIM# 173910) genes, encoding polycystins 1 or 2, lead to the dysfunction of the polycystin complex, contributing to the vast majority of all cases of ADPKD [8]. These mutations result in a decrease in intracellular calcium concentration, thereby increasing the activity of adenylyl cyclase as well as cyclic adenosine monophosphate (cAMP) [9]. The sustained high levels of intracellular cAMP in the proximal, distal, and collecting ducts leads to abnormal proliferation of tubular epithelial cells. Chloride-controlled fluid secretion and cAMP production are two of the main components of cyst formation and growth [10,11,12]. Uncontrolled cyst growth can result in the displacement of adjacent nephrons, the destruction of the renal parenchyma, and, eventually, the enlargement of the kidneys and progressive renal failure [13,14]. A notable feature of ADPKD is its broad and significant inter- and intra-familial variability. There is a range of sequence defects in the PKD genes, which could partially explain the variability in disease severity and progression [15,16]. Additionally, the course of the disease may also be influenced by random somatic genetic events and modifier genes [17,18]. However, the identification of such modifier genes, which could have crucial therapeutic implications, remains challenging in humans [19]. 

Nephronophthisis (NPHP), an autosomal recessive cystic kidney disease, stands as the most prevalent genetic cause of ESRD during the first three decades of life [20,21]. The estimated incidence of new cases ranges from 1 in 50,000 to 1 in 1,000,000 live births. NPHP in three clinical forms—infantile, juvenile, and adolescent—is distinguished based on the onset of ESRD at ages 1, 13, and 19, respectively [22,23,24]. Renal histology reveals a distinctive triad of tubular basement membrane disruption, tubulointerstitial nephropathy, and corticomedullary cysts [25]. Renal size is within normal range or slightly reduced, except in infantile NPHP type 2, where moderate renal enlargement may occur. This contrasts with ADPKD, where cysts are evenly distributed throughout the organ and result in severe renal enlargement. NPHP is associated with various clinical manifestations such as tapeto-retinal degeneration (Senior–Løken syndrome), cerebellar vermis aplasia (Joubert syndrome), Cogan-type oculomotor apraxia, mental retardation, liver fibrosis, or cone-shaped epiphyses of the phalanges. Infantile NPHP type 2 may be associated with situs inversus, retinitis pigmentosa, or cardiac ventricular septal defect [26,27]. Although most NPHP gene products (NPHPs) are localized in the primary cilium, the precise molecular functions of NPHPs remain largely unknown [28,29].

## 2. The *Han:SPRD* Rat Model

The intricacy of genetic studies of ADPKD in humans has led to the use of rodent models of spontaneous polycystic kidney disease (PKD) to characterize mechanisms involved in renal cystogenesis [30]. The *Han:SPRD* rat model stands as the sole well-documented animal model of inherited PKD with an autosomal dominant pattern of inheritance, closely mirroring several features of human ADPKD, including renal hyperplasia, azotemia, and extrarenal manifestations (Table 1) [31]. This model was first described by Dr. Deerberg as a spontaneous derived model (Central Institute for Laboratory Animal Breeding, Hannover, Germany) in 1989 [32]. Subsequently, the colony of inbred *Han:SPRD* rats was established in the laboratory in Mannheim, under the control of Dr. Gretz after 18 additional generations of inbreeding, and named *PKD/Mhm* (University of Heidelberg, Mannheim, Germany) rats [33,34,35]. A striking contrast in the severity of cystic disease is evident when comparing homozygous and heterozygous affected animals (Table 2). Homozygous rats developed massive renal enlargement and marked azotemia, and died at approximately 3 weeks of age. Clinically, affected homozygous rats show a markedly distended abdomen and, in post-mortem analysis, reveal enlarged kidneys, often comprising more than 20% of total body weight [36]. At 9 weeks of age, all heterozygous animals exhibit kidney enlargement and abnormal ultrasound features. Male rats, at this age, commonly display hyperechoic renal cortex with cyst formation and loss of corticomedullary junction. On the contrary, female rats have no visible cysts at this age, and enlarged hyperechoic renal cortex is the most prevalent abnormality. However, at 12 weeks of age, cysts are ultrasonographically visible in female rats as well [37]. By 6 months of age, in male heterozygotes, a slow progression of cystic kidney disease with interstitial fibrosis is observed. Clinical symptoms and signs in males include weight loss, polyuria, polydipsia, and eventually end-stage renal failure and death [38]. Female heterozygotes develop slowly progressive cystic kidney disease but no interstitial fibrosis or azotemia. Furthermore, while female rats have a normal average blood pressure, PKD male rats at 6 months of age exhibit markedly increased blood pressure. Death in the first year of life is extremely uncommon; however, during the second year, the survival rate of male rats decreases rapidly, as 50% of the animals die [32].

## 3. Kidney Histology of *Han:SPRD* Rats

The histological differences between homozygous (Cy/Cy) and heterozygous (Cy/+) *Han:SPRD* rats are apparent [50]. In homozygotes, cystic dilatation affects all segments of the nephron except the glomeruli [51]. However, the histology results in heterozygotes differ according to the age and sex of the rats. Early on, Cy/+ rats have minimal renal cyst development, mainly on the proximal tubules of juxtamedullary nephrons [39]. At 5 to 6 weeks, cysts become more prominent but still mainly affect the proximal tubules. By 8 to 10 weeks cysts fill the entire organ, with male kidneys more often affected than females (Figure 1) [38]. Increased cell proliferation is crucial in PKD, leading to abnormally large renal tubules. These tubules, normally made up of only a few cells, grow several millimeters in diameter and have a large number of cells in the cyst walls. Cysts are primarily found in the inner cortex, while they are not as prevalent in the outer cortex. A few dilated tubules extend to the outer medulla but not to the inner medulla. The epithelial cells lining the cyst exhibit various degrees of morphological immaturity. Most cysts exhibit varying degrees of degradation of the proximal tubular brush border. Transitions are frequently observed from zones with an intact brush border to areas where the cells are polygonal, and they rarely have brush border. It is unclear why basilar cyst epithelia lose their differentiation. However, considering the interrelation between differentiation and proliferation, these changes could be attributed to genetic diseases that dysregulate cell proliferation [52]. The cyst walls also exhibit basement membrane thickening, particularly in areas of cellular immaturity, suggesting a relationship between basement membrane thickening and cellular dedifferentiation. Thickened basement membranes are associated with increased immunoreactivity for collagen type IV, laminin, and fibronectin. Inflammatory cells are focally present in the peri-cystic interstitium, and there is a significant increase in interstitial fibrosis and the number of inflammatory cells [36,53]. 

## 4. Genetics

The phenotypic ratios observed from mattings of rats affected by PKD, as well as from mattings of healthy and diseased rats, align consistently with those of an autosomal dominant trait transmitted by a single gene [50]. In an experimental backcross population comprising affected *Han:SPRD* rats and unaffected Wistar Ottawa Karlsburg rats, a comprehensive genome scan employing 117 microsatellite markers successfully identified and allowed the genetic dissection of PKD on rat chromosome 5. In a detailed linkage mapping of rat chromosome 5, the PKD locus is located approximately 25 cM from the proenkephalin gene [33]. This region serves as a quantitative trait locus that controls PKD, kidney mass, and plasma urea concentration. The gene that induces PKD in the *Han:SPRD* rat was neither *PKD1*, which is located on human chromosome 16, nor *PKD2*, which is located on human chromosome 4. Therefore, Bihoreau et al. denoted a new locus as *PKDR1*. Searching the EST database for similar sequences to rat Pkdr1, Kaisaki et al. pinpointed human *ANKS6*, located on human chromosome 9, as the human *PKDR1* [52]. The *ANKS6* gene contains 15 coding exons and spans 64.5 kb [50]. The deduced protein, which is called samcystin, was found to be expressed specifically in the proximal renal tubules, and it contains 11 tandem ankyrin repeats in the NH2 terminus and a sterile alpha motif (SAM) in the COOH terminus. Positional cloning and mutational analysis revealed a cytosine-to-thymine transition, resulting in an arginine-to-tryptophan substitution at amino acid 823 in the protein sequence. The arginine residue mutated in the rat sequence of the SAM domain, suggesting an important functional role. The removal of arginine and insertion of tryptophan disrupt the overall tertiary structure of the protein and prevent the mid-loop surface of the ANKS6–SAM domain from adopting a complementary fold for binding to RNA and other proteins, including the ANKS3 and other signaling molecules, that are critical to its function [31,51,54]. Among the proteome in humans, *Drosophila*, and *Caenorhabditis elegans*, which contain a SAM domain C-terminal and ANK repeats, only six of twenty-one lack arginine at this specific position, and, in just two of the six cases, the residue is hydrophobic, while no occurrences of tryptophan were identified [39,51,52]. A more detailed characterization of this protein, which lacks any structural similarities with known polycystins, may provide new insights into the pathophysiology of renal cyst development in human patients. The large number of abnormally expressed genes in each of the polycystic kidney disease models suggests that the responsible defect most likely affects an early step within the same cascade of events—perhaps an abnormal signaling or transcription factor cascade, which triggers afterwards the same abnormal cascade of gene expression events [53]. This affects a variety of genes and eventually leads to cells that function abnormally and exhibit the same cystic phenotype. Although caused by different genetic defects, they have phenotypically similar alterations in cell differentiation, growth hormone and oncogene expression, metabolic transport, glucocorticoid metabolism, and fatty acid and lipid composition. Thus, the often cited pathogenetic features of cystic disease (increased cell proliferation and fluid accumulation) cannot be studied as independent features, but must be considered as links in the same complex network of intracellular events that might even influence each other [33].

## 5. Localization and Function of Samcystin in Kidney

Early postnatal kidney samcystin expression was observed in wild-type rats during the final stage of renal development, with levels decreasing from 7 to 45 days. In the kidneys of heterozygous and homozygous rat kidney (*Han:SPRD-Cy/+*; or *Han:SPRD-Cy/Cy*), samcystin expression was downregulated at 3 and 7 days, and then was markedly increased compared to age-matched normal kidneys [31]. Immunohistochemical analysis revealed that samcystin was distributed on the brush border of proximal tubules in normal rat kidneys. Conversely, in Cy/+ kidneys, robust samcystin staining was observed in cyst-lining epithelial cells, accompanied by loss of apical localization and increased numbers of proliferating cell nuclear antigen-positive cells in cyst-lining epithelia (Figure 2) [31]. *Anks6* is exclusively expressed in the proximal tubules of the adult kidney, thus indicating that cysts primarily originate from these tubules in mutants [39]. While *anks6* (p.R823W) may be critical for the maintenance of proximal tubular function, its mild expression in other tubular segments/tubules may lead to cyst formation, albeit to a lesser extent and in older age. By inhibiting or impairing the ability of the cell to proliferate, Anks6 disrupts the delicate balance between the opposing processes of cell proliferation and apoptosis. Furthermore, in heterozygous neonatal rats *Han:SPRD* (Cy/+), distinct mRNA expression of *anks6* (p. R823W) was observed in the tubular epithelium and podocytes. Therefore, the proteinuria of some patients could be attributed to *ANKS6* mutation, which may indicate a critical function of ANKS6 not only in tubular epithelial cells but in podocytes as well [55]. Finally, molecular analysis indicates that anks6 localizes in the primary cilium of tubular epithelial cells, particularly in the inversin compartment (IC), where interacts with colocalized cystoproteins INVS, NHPH3, and NEK8 (Figure 3), [47,55,56]. Studies using human RPE1 cells, a classic model for studying primary cilia in vitro, have shown that knocking out *anks6* resulted in shortened or duplex cilia, suggesting a regulatory role of anks6 in cilia formation [57].

## 6. Role of Samcystin in Cystogenesis

Samcystin expression is consistent with the observed development of cysts in heterozygous rats, as well as the widespread cystic growth and renal enlargement in homozygous rats. The main sites of its expression were the proximal tubules, with particular emphasis on the brush border as the central point of expression [38]. Overexpression of samcystin in mutants resulted in a loss of specificity for its localization to the brush border in cyst-lined epithelial cells [58], indicating a link between the Cy mutation in the SAM domain and mislocalization of the protein. Overall, a point mutation of *anks6*, which results in the substitution of the amino acid “R823W” in the samcystin domain, results in the misexpression and mislocalisation of samcystin in the cystic epithelial cell [31]. The Cy mutation may interfere with samcystin’s interactions with other signaling molecules, while also increasing samcystin levels in response to the loss of a functional protein. This could lead to the accumulation of inadequate samcystin proteins and a dominant negative reaction, resulting in cellular differentiation and cyst formation [47]. Though the precise role of *ANKS6* in renal cystogenesis is not fully understood, a recent study provides a compelling argument for the potential role of *ANKS6* and *NEK8* in regulating polycystic kidney function. The authors suggest a mechanistic connection between the IC and the phenotypic results of both *ANKS6* and *NEK8* polycystic kidney mutants. ANKS6 transports NEK8 from the cytoplasm to the cilia, where it is phosphorylated in the presence of INV and NPHP3, suggesting its role as both a substrate and a functional activator of NEK8 kinase activity [59]. NEK8 has already been identified as an important regulator in kidney and liver cystogenesis and seems to be involved in the same signaling cascade as *PKD1* and *PKD2* [60,61]. Moreover, a study by Nakajima et al. has demonstrated that the IC functions as an intraciliary signal-generating center and that ANKS6 may be involved in the transmission of signals from cilia to the cytoplasm [62], possibly indicating that the phosphorylation status of ANKS6 influences the stability of the protein [47,62,63]. The idea that ANKS6 interacts with proteins that are pivotal to normal kidney development is intriguing and supported by numerous pieces of evidence. The altered B-Raf/MEK/ERK, AKT/mTOR, and RXR pathways in Cy/+ kidneys suggest that ANKS6 may be involved in cystogenesis, although its exact role is unknown [64]. Other studies have identified ANKS6, ankyrin repeat, and SAM-domain containing protein 3 (ANKS3) as potential ligand partners [65]. The R823W mutations adversely impact the structural integrity of the SAM domain within the ANKS6 domain, affecting the binding to the SAM domain of ANKS3. This structural insight sheds light on the defect observed in *HAN:SPRD (Cy/+)* rats, suggesting a novel pathophysiology of cystic disease through the SAM domain (ANKS3) [66]. Furthermore, array-based gene expression profiling in renal tissue of adult *HAN:SPRD (Cy/+)* rats revealed altered aqp3/aqp4 transcriptional regulation, indicating that impaired ANKS6–ANKS3 binding interferes with vasopressin-mediated pathways, which are either directly or indirectly associated with polycystic kidney disease. Renal aqp2 expression is also stimulated in very early stages of PKD in young (10 days old) *HAN:SPRD (Cy/+)* rats, indicating that increased aqp2 activity in response to experimental anks3 downregulation could be the cause of priming molecules in cystogenesis. Additionally, evidence of BICC1-mediated regulation of renal cAMP signaling in bicc1^−/−^ mutant mice suggests a biological interaction between BICC1, ANKS6, and ANKS3 in regulating vasopressin-/cAMP-signaling pathways. Renal gene transcription profile data suggest that vasopressin-mediated mechanisms may be a common denominator in the pathophysiology of PKD [66,67]. Finally, although samcystin is structurally distinct from the polycystic kidney disease-related proteins, polycystins 1 and 2, it cannot be overlooked that samcystin’s SAM domain plays a direct role in polycystin-1-related signaling pathways [38,47,68].

## 7. Extrarenal Manifestations of ADPKD 

Cerebrovascular aneurysms, heart valve abnormalities, and colonic diversions are the most frequent extrarenal manifestations of ADPKD. Notably, *Han:SPRD (Cy/+)* rats lack cerebral aneurysms, in contrast to humans. In the heart, the splice-site and truncating mutations of *ANKS6* were linked to hypertrophic obstructive cardiomyopathy, aortic stenosis, pulmonary stenosis, patent ductus arteriosus, and situs inversus [47]. Furthermore, male rats exhibit fibrosis, ranging from isolated interstitial connective tissue collections to dense collagen in regions devoted to normal cardiac fibers [32]. The involvement of *ANKS6* in cardiac development and function is further supported by the observation that a patient who had a truncation in *ANKS6* at the N-terminal end of the SAM domain, who also presented aortic stenosis, resulted in obstructive cardiomyopathy [69]. Regarding renal osteodystrophy, impaired kidney function in male rats leads to renal osteodystrophy, with conspicuous hyperplasia of the parathyroid glands and replacement of bone by fibrous tissue [70]. Furthermore, metastatic calcification affects several organs, including the media wall of large arteries such as the aortic arch, thoracic aorta, abdominal aorta, and renal artery, which are calcified. Focal mineralization of heart muscle fibers occurs as well [71,72]. Regarding enteritis, uremic enteritis is observed in about 40% of male *HAN:SPRD* rats. Macroscopically, the calcification of the gastric glands, the connective tissue, and the muscular layer of the forestomach appears as white streaks seen macroscopically in the stomach wall. Additionally, focal wall edema and modest polymorphonuclear granulocyte infiltration were present in gastric, colon, and cecum walls in male rats. In 6-month-old heterozygous males, thickening of the colon and cecum walls with intramural bleeding is common [51]. In the liver, although male rats exhibit more severe disease, liver cysts are more common in affected female rats. In a previous study, 42% of heterozygous female *Han:SPRD (Cy/+)* rats developed liver cysts, compared to only 3% of homozygous unaffected rats of the same age (*p* < 0.001). Notably, no relationship was found between the number of pregnancies and liver cyst formation. Interestingly, female rats with liver cysts were older than 1 year and appeared to have greater levels of uremia [35]. The liver cysts observed in murine models, with PKD resembling human liver cysts in two aspects: firstly, they were only found in later stages, and, secondly, the cyst epithelium resembled the epithelium in biliary ducts. Expression studies and biochemical investigations in Anks6 knockout mice demonstrate that the dysregulation of YAP transcriptional activity in the bile duct-lining epithelial cells is the cause of biliary defects in the anks6-deficient liver [73]. According to this study, ANKS6 inhibits the Hippo pathway effector proteins YAP1, TAZ, and TEAD4 transcriptional activity, to block Hippo signaling in the liver during the formation of bile ducts [73,74]. Histological analyses show mild periportal fibrosis, inflammatory infiltrations (primarily in the connective tissue of the portal locations [35]), a sharp reduction in the thickness of the periportal mesenchyme, and a complete absence of peribiliary fibroblasts, indicating that the abnormalities in portal morphogenesis may be caused by impaired crosstalk between the biliary epithelium and the portal mesenchyme [73]. These findings in humans and in an orthologous anks6 knockout mouse model offer compelling evidence in support of the hypothesis that CHF is caused by the activation of pro-inflammatory signaling in the periportal space by activated bile duct epithelial cells [75]. Moreover, cysts have also been described in humans with ADPKD in other organs including the pancreas, lungs, spleen, ovaries, testes, epididymis, thyroid, uterus, broad ligament, and bladder. Comparable cystic alterations could not be identified in any of these organs in *Han:SPRD* animals. 

## 8. Preclinical Trials

Many studies in the bibliography have used *Han:SPRD* rats as a model for PKD. Table 3 illustrates the most significant studies that have been conducted in *Han:SPRD*; however, in clinical studies, the results have been controversial. Among the most noteworthy clinical studies in PKD is the use of mTOR inhibitors as a potential treatment to prevent cyst onset, expansion, and PKD progression. The mTOR pathway, regulating a vast array of cellular functions and promoting cell growth and proliferation, was targeted in a sirolimus trial, proving highly effective in *Han:SPRD* rats. This trial showed decreased phosphorylation of S6K and reduced PKD progression, indicating the crucial role of the mTOR pathway in the pathogenesis of cyst growth in *Han:SPRD*. Long-term usage of rapamycin/sirolimus in PKD leads to a marked reduction in CKD progression, hypertension, and cardiac hypertrophy, with minimal side effects [76,77]. Additional studies are necessary to determine if prolonging sirolimus or other mTOR inhibitors’ treatment can decrease mortality in *Han:SPRD* rats. 

Renal disease progression in PKD is a complex process that involves various factors and mechanisms. Two crucial aspects highlighted by studies are the inhibition of COX-2 and the use of calcium channel blockers in improving renal disease progression. COX-2 inhibitors, such as NS-398, have shown renoprotective effects in studies conducted on *Han:SPRD* rats with chronic renal injury [78]. These inhibitors—due to anti-inflammatory, antiproliferative, and antifibrotic effects—contribute to the amelioration of renal injury. In particular, selective COX-2 inhibition by NS-398 has demonstrated a significant reduction in cyst growth and interstitial fibrosis in *Han:SPRD* rats. Although these histological changes are not accompanied by improvements in renal function, they are likely to provide long-term benefits by slowing down disease progression. Further investigation is needed to examine the effects of different doses and longer-term use of selective COX-2 inhibitors on cystic renal disease progression [78]. Additionally, the use of calcium channel blockers, such as verapamil, has also been explored in *Han:SPRD* rats [79]. Due to lower steady-state intracellular calcium levels in cystic cells relative to normal cells, cyclic AMP (cAMP) is mitogenic in cystic but antimitogenic in normal human kidney cells. Verapamil-induced inhibition of intracellular calcium entry led to cAMP-dependent cell proliferation in normal renal cells. The increased cell proliferation and apoptosis, along with elevated expression and phosphorylation, of B-Raf were observed in verapamil-treated Cy/+ rats. This coincided with the stimulation of the mitogen-activated protein kinase MEK/ERK (mitogen-activated protein kinase kinase/extracellular-regulated kinase) pathway. These cellular changes were linked to the elevated kidney weight and cyst index in verapamil-treated Cy/+ rats. Verapamil had no impact on B-Raf stimulation or kidney morphology in wild-type rats. In conclusion, treatment of Cy/+ rats with calcium channel blockers increases activity of the B-Raf/MEK/ERK pathway, increasing cyst development in the presence of endogenous cAMP, aggravating renal cystic disease [79]. 

As part of the development of new treatment options for slowing chronic kidney disease progression, sodium–glucose transport protein 2 inhibitors (SGLT2i) have also been tested in patients with polycystic kidney disease. In polycystic kidneys, the inhibition of SGLTs was linked to a reduction in ERK1/2 phosphorylation and tubular epithelial cell proliferation. Phlorizin, a dual SGLT1/2 inhibitor, demonstrated efficiency in minimizing cyst formation and halting the deterioration of renal function in the *Han:SPRD* rat [80]. Since much of the glucose is reabsorbed by the uncontrolled SGLT1 in the proximal tubule, dapagliflozin, an SGLT2 inhibitor, was a less effective glucosuric drug than phlorizin and it did not reduce cyst formation in the *Han:SPRD* rat. Despite the fact that dapagliflozin enhanced renal function in *Han:SPRD* rats, no evidence of hyperfiltration was discovered, particularly since albuminuria was reduced. Collectively, the above data imply that dapagliflozin has PKD model-specific effects, but its impact on human ADPKD may be unpredictable [81,82,83]. 

As mentioned earlier, cardiovascular complications are the most common cause of death in PKD patients. In the early stages of the disease, the *Han:SPRD* rat experiences diastolic dysfunction, which may subsequently change to systolic dysfunction. Fish oil (FO) reduced diastolic dysfunction without changing blood pressure, suggesting that it may have a cardioprotective impact on the heart, independent of the hemodynamic stress [84]. The positive benefits of FO in diseased rats may be related to a decrease in cardiac fibrosis, a significant factor in the emergence of diastolic heart failure. In summary, the *Han:SPRD* rat model is suitable for investigating chronic renocardiac syndrome, given that diseased rats exhibit larger hearts, higher blood pressure, and altered systolic and diastolic functions compared to normal rats. 

**Table 3 biomedicines-12-00362-t003:** Comparison of clinical and preclinical studies with *Han:SPRD* as a polycystic kidney disease animal model.

Mechanism of Action	Intervention	*Han:SPRD*		Human	
		Cyst Growth	Other Effect	Cyst Growth	Other Effect
mTOR inhibitor	Sirolimus	Reduction [76,77]	Reduction of CKD progression [76,77]	Reduction [85]	No effect in CKD progression [85]
COX-2 inhibitor	NS-398	Reduction [78]	N/A	N/A	N/A
SGLT1,2i	Phlorizin	Reduction [80]	Reduction of CKD progression [80]	N/A	N/A
SGLT2i	Dapagliflozin	No effect [81,82,83]	Reduction of CKD progression [81,82,83]	Increase [86]	Increase of CKD progression [86]
Calcium channel inhibitors	Verapamil	Increase [79]	Increase of CKD progression [79]	N/A [87]	Increase of CKD progression [87]
Natural vitamin	Fish oil	None [84]	Reduction of diastolic dysfunction [84]	N/A	N/A

CKD: chronic kidney disease, mTOR: Mammalian target of rapamycin, COX-2: cyclooxygenase-2, SGLT2i: Sodium–glucose co-transporter 2 inhibitor, N/A: non-applicable.

## 9. ANKS6 in Humans

Recent findings have established an association between *ANKS6* and human kidney disease, particularly with nephronophthisis. The genomic sequence of *PKDR1* on chromosome 9q22.33 spans approximately 64.5 Mb and includes 15 coding exons and one non-coding exon. The gene responsible for encoding human samcystin has been sequenced, revealing new coding and non-coding polymorphisms [33]. The UniProt database lists four isoforms of *ANKS6*, denoted as Q68DC2-1 to Q68DC2-4, although experimental confirmation is lacking for two of them [52]. Notably, many of the mutations detected are located in the ankyrin repeats, with a specific truncation at tyrosine790 observed in the SAM domain. This truncation significantly impacts normal function of anks6 in both rats and humans. Mutation analysis of nephronophthisis cohorts have identified patients with truncating, splice-site, and non-synonymous missense mutations [52,69]. Individuals in these cohorts generally exhibit early, infantile onset of PKD, except for those in family NPH316, where juvenile onset is observed [47]. Patients with missense mutations showed cystic kidney disease without extrarenal manifestations. On the other hand, truncating mutations were linked to an enlarged renal size, PKD, and early onset ESRD, as well as severe extrarenal defects [88]. The severity of the phenotype in patients with splicing mutations is linked to the location of the causative mutations. Different conformational changes in ANKS6 and/or defective interactions with different interacting partners may be the cause of these different phenotypic characteristics [89].

## 10. Conclusions

In summary, the development of ADPKD in *Han:SPRD* rats can be attributed to a recessive mutation in the *anks6* gene, located on chromosome 5. The anks6 protein, samcystin, localizes to the ciliary base and forms extensive complexes with other cysto-proteins such as INVS, NPH3, and NEK8. While the precise molecular function of anks6 remains elusive, accumulating evidence suggests its crucial role not only in maintaining the structural integrity of primary cilia, but also in cell signaling, proliferation, and DNA damage response. ANKS6 is a key player in the NPHP module, providing insights into the observed phenotypic overlap with abnormalities in the heart and liver observed in patients with single mutations in this gene. Notably, mutations in *ANKS6* have also been found in humans, leading to cystic kidney disease. Consequently, the *Han:SPRD* rat model emerges as a valuable tool for studying the pathogenesis of polycystic kidney disease and evaluating potential therapeutic interventions.

## Figures and Tables

**Figure 1 biomedicines-12-00362-f001:**
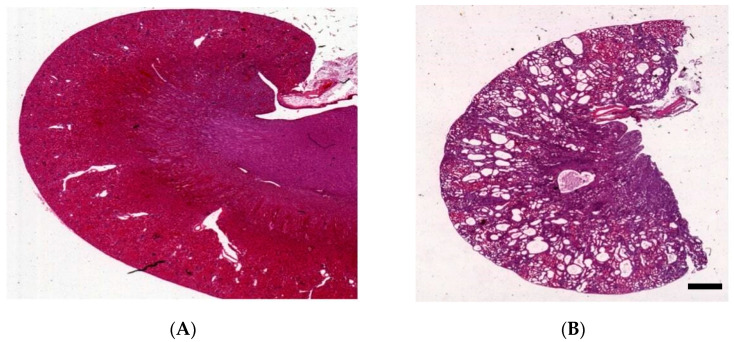
Characteristics of the kidney in wild-type (**A**) and *Han:SPRD* Cy/+ rats (**B**), hematoxylin and eosin (HE)-stained kidneys of 12-week-old rats (unpublished results, scale bar: 1.5 mm).

**Figure 2 biomedicines-12-00362-f002:**
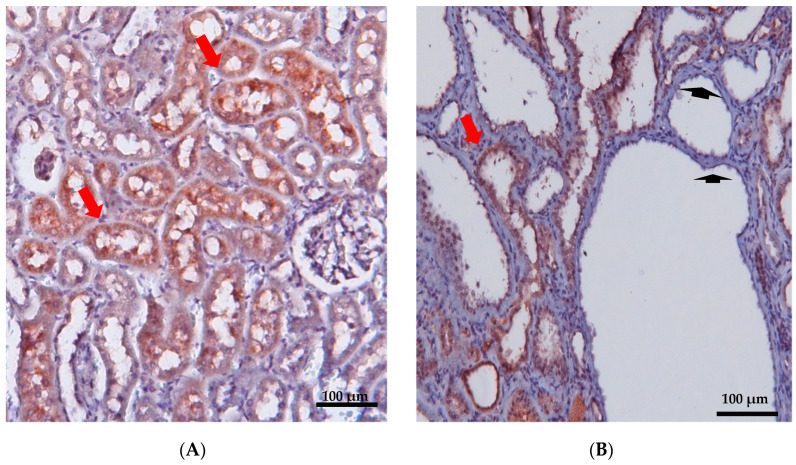
Immunohistochemical staining of renal cortex was performed in 12-week-old wild-type (**A**) and *PKD/Mhm (Cy/+)* (**B**) rats for Anks6. Anks6 localized in the brush border of proximal tubules (red arrow). Cysts came mainly from proximal tubules, but there were also cysts without anks6 expression (arrowhead) (unpublished results, the Anti-Anks6 sc-515124 Ab Santa Cruz Biotechnology, was used in a 1:150 dilution; scale bar 100 μm).

**Figure 3 biomedicines-12-00362-f003:**
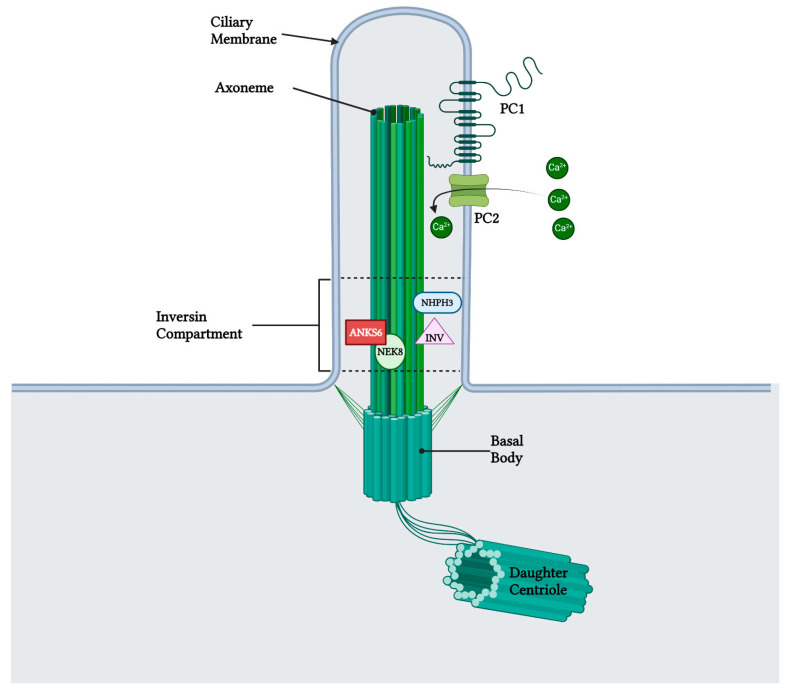
Anks6 localizes in the inversin compartment of renal primary cilium and interacts with other nephrocystins in transition zone. The polycystin complex, which acts as a calcium channel, is localized in the primary cilia as well, however a clear interaction with anks6 has not been yet revealed. “Created with BioRender.com, accessed on 26 January 2024”.

**Table 1 biomedicines-12-00362-t001:** Comparison of renal and extrarenal manifestations in *Han:SPRD* rats and humans with ADPKD.

Manifestation	*Han:SPRD*	Human
Renal cysts	Proximal ducts [39]	Distal and collecting ducts [40,41]
Liver cysts	40% of female [35]	50% of male and female [42]
Pancreas and spleen	None [35]	Cysts [43]
Brain	None [35]	Cerebrovascular aneyrysms [44,45]
Bones	Renal osteodystrophy [32]	Renal osteodystrophy [46]
Heart	Cardiomyopathy [47]	Cardiomyopathy [48,49]
	Valvular defects Situs inversus	Valvular defects Arrythmias

**Table 2 biomedicines-12-00362-t002:** Clinical and ultrasound findings in *Han:SPRD* rats.

Genotype	Age				
	3 weeks	9 weeks	12 weeks	6 months	>1 year
Male Cy/+	Mild cysts	Multiple cysts	Kidney enlargement	CKD	Nortality > 50%
Female Cy/+	No cysts	No cysts	Mild cysts	Kidney enlargement	CKD
Cy/Cy	ESRD Mortality 100%	-	-	-	-

CKD: chronic kidney disease, ESRD: end stage renal disease, Cy/+: heterozygotes, Cy/Cy: homozygotes.

## Data Availability

Data are available upon request.

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
