# Peer review of "The Han:SPRD Rat: A Preclinical Model of Polycystic Kidney Disease"

_biomedicines, 2024, doi:10.3390/biomedicines12020362_

Round 1

Reviewer 1 Report

Comments and Suggestions for Authors

The authors have described usefulness of Han:SPRD rat as a polycystic kidney disease model in this review paper, containing history, genetics, histopathology, cyst formation mechanisms, extrarenal manifestation, and clinical relevance for the Han:SPRD rats. The contents are interesting because there are hints concerning the therapeutics of polycystic kidney diseases. This referee has some cosmetic points.

Comments

(1) Table 2: This referee recommends the authors to separately describe the effect of each drug on cyst formation and other injuries. 

(2) This referee would like the authors to write the manuscript with common nouns to be standardized in sentences, starting with a lowercase letter, including samcystin and drug names.

Comments on the Quality of English Language

This referee would like the authors to write the manuscript with common nouns to be standardized in sentences, starting with a lowercase letter, including samcystin, drug names, and so on.

Author Response

Response to Reviewer 1

We would like to thank reviewer 1 for his/her insightful comments.

Point 1: Table 2: This referee recommends the authors to separately describe the effect of each drug on cyst formation and other injuries. 

Response 1: Two additional columns have been added in table 2 which descreibe the effect of each drug on cyst formation and other injuries.  according to Reviewer’s 1 suggestion

Mechanism of

action

Intervention

Han:SPRD

Human

Cyst growth

Other effect

Cyst growth

Other effect

mTOR inhibitor

Sirolimus

Reduction [73,74]

Reduction of CKD progression [73,74]

Reduction [82]

No effect in CKD progression [82]

COX-2 inhibitor

NS-398

Reduction [75]

N/A

N/A

N/A

SGLT1,2i

Phlorizin

Reduction [77]

Reduction of CKD progression [77]

N/A

N/A

SGLT2i

Dapagliflozin

No effect [78–80]

Reduction of CKD progression [78–80]

Increase [83]

Increase of CKD progression [83]

Calcium channel inhibitors

Verapamil

Increase [76]

N/A

N/A

Increase of CKD progression[84]

Natural vitamin

Fish oil

None

Reduction of diastolic dysfunction [81]

N/A

N/A

CKD; chronic kidney disease, mTOR; Mammalian target of rapamycin, COX-2; cyclooxygenase-2, SGLTi; Sodium-glucose trasporter inhibitor, N/A; non applicable.

Point 2: This referee would like the authors to write the manuscript with common nouns to be standardized in sentences, starting with a lowercase letter, including samcystin and drug names

Response 2: Changes have been made accordingly to Reviewer’s 1 suggestions. (e.g. samcystin, sirolimus, anks6)

Reviewer 2 Report

Comments and Suggestions for Authors

Thoughtful, thorough review and no major criticism.

Author Response

We would like to thank reviewer 2 for reviewing our manuscript.

Reviewer 3 Report

Comments and Suggestions for Authors

Congratulations for a well written article. I think the following article should also be included in the review, making a valuable addition to it: "Kugita M, Nishii K, Morita M, Yoshihara D, Kowa-Sugiyama H, Yamada K, Yamaguchi T, Wallace DP, Calvet JP, Kurahashi H, Nagao S. Global gene expression profiling in early-stage polycystic kidney disease in the Han:SPRD Cy rat identifies a role for RXR signaling. Am J Physiol Renal Physiol. 2011 Jan;300(1):F177-88. doi: 10.1152/ajprenal.00470.2010. Epub 2010 Oct 6. PMID: 20926632."

Comments on the Quality of English Language

Some minor corrections and rephrasing should be performed (e. g. "In homozygotes, cystic dilatation affects all segments of the nephron except for the glomeruli [40]. While in heterozygotes, the histological findings vary depending on rat age and sex.").

Author Response

Response to Reviewer 3

We would like to thank reviewer 3 for his/her insightful comments.

Point 1: Congratulations for a well written article. I think the following article should also be included in the review, making a valuable addition to it: "Kugita M, Nishii K, Morita M, Yoshihara D, Kowa-Sugiyama H, Yamada K, Yamaguchi T, Wallace DP, Calvet JP, Kurahashi H, Nagao S. Global gene expression profiling in early-stage polycystic kidney disease in the Han:SPRD Cy rat identifies a role for RXR signaling. Am J Physiol Renal Physiol. 2011 Jan;300(1):F177-88. doi: 10.1152/ajprenal.00470.2010. Epub 2010 Oct 6. PMID: 20926632."

Response 1: The above mentioned article has been included in section 6. Role of Samcystin in cystogenesis, as follows: “…normal kidney development is intriguing and supported by numerous pieces of evidence. The altered B-Raf/MEK/ERK, AKT/mTOR, and RXR pathways in Cy/+ kidneys suggest that ANKS6 may be involved in cystogenesis, although its exact role is unknown [53]

Point 2: Some minor corrections and rephrasing should be performed (e. g. "In homozygotes, cystic dilatation affects all segments of the nephron except for the glomeruli [40]. While in heterozygotes, the histological findings vary depending on rat age and sex.").

Response 2 Rephrasing has been performed in the following parts:

 Page 3, lines 126-128 “In homozygotes, cystic dilatation affects all segments of the nephron except the glomeruli [40]. However, the histology results in heterozygotes differ according to the age and sex of the rats.”

Page 4, lines 212-215 “Accordingly, the often cited pathogenic features (increased cell proliferation and fluid accumulation) of cystic disease cannot be considered independently of each other, but rather as part of a complex network of intracellular events that may even influence one another. [33].”

Page 8, lines 686-688 As part of the development of new treatment options for slowing chronic kidney disease progression, sodium-glucose transport protein 2 inhibitors (SGLT2) have also been tested in patients with polycystic kidney disease.”

Reviewer 4 Report

Comments and Suggestions for Authors

Dear Authors,

This is an interesting a well written article. The manuscript entitled: The Han:SPRD rat: a preclinical model of polycystic kidney disease. The MS is mainly focused to explore the utility of Han:SPRD  rat model, highlighting its phenotypic similarity to human ADPKD. Specifically, the authors discuss its role in preclinical trials and its importance for investigating the pathogenesis of the disease and developing new therapeutic approaches. However there a minor concerning aspects regarding the manuscript.

The MS is very difficult to global follow by the huge  amount of text in every point I will thank to the authors to reduce the text, trying to incorporate  figures or tables to show part of the results. In this sense, a schematic  figure with protein ways involved in your hypothesis will be appreciate.

.-Please correct italic for genes PKD1 and PKD2 within the text.

.-Some references may help to complete  the article:

Claus LR et al., 2023. kidney International

Notoli et al., 2008

Comments on the Quality of English Language

No further comments

Author Response

Response to Reviewer 4

We would like to thank reviewer 4 for his/her insightful comments.

Point 1: The MS is very difficult to global follow by the huge  amount of text in every point I will thank to the authors to reduce the text, trying to incorporate  figures or tables to show part of the results. In this sense, a schematic  figure with protein ways involved in your hypothesis will be appreciate.

Response 1: An additional table has been created that summarizes the cilical and ultrasound findings of HAN:SPRD at different ages. We also designed a figure which depicts the localization of anks6 in the primary cilia. The table and figure have been added in the respective parts of the manuscript.

Table 2. Clinical and ultrasound findings of HAN:SPRD rats.

Genotype

Age

3 weeks

9 weeks

12 weeks

6 months

>1 year

Male Cy/+

Mild cysts

Multiple cysts

Kidney enlargement

CKD

Nortality> 50%

Female Cy/+

No cysts

No cysts

Mild cysts

Kidney enlargement

CKD

Cy/Cy

Kidney enlargement

ESRD

-

-

-

-

CKD: chronic kidney disease, ESRD: end stage renal disease, Cy/+: heterozygotes, Cy/Cy: homozygotes

Figure 3. Anks6 localizes in the inversin compartment of renal primary cilium and interacts with other nephrocystins in transition zone. “Created with BioRender.com”

Point 2: Please correct italic for genes PKD1 and PKD2 within the text.

Response 2: PKD1 and PKD2 have been changed to Italic.

Point 3: Some references may help to complete  the article:

Claus LR et al., 2023. kidney International

Notoli et al., 2008

Response 3: The references have been added in the section 6. Role of Samcystin in cystogenesis, page 5, lanes 230-232 “NEK8 has been already identified as an important regulator in kidney and liver cystogenesis and seems to be involved in the same signaling cascade with the PKD1 and PKD2 [51,52].”